# Time to care: why the humanities and the social sciences belong in the science of health

Brendan Clarke,[1] Virginia Ghiara,[2] Federica Russo [3]

[1]Department of Science & Technology Studies, University College London, London, United Kingdom
[2]Department of Philosophy, University of Kent, Canterbury, Kent, UK
[3]Department of Philosophy, Universiteit van Amsterdam, Amsterdam, The Netherlands

**Correspondence to**
Dr Federica Russo;
federica.russo@gmail.com

## ABSTRACT

Health is *more* than the absence of disease. It is also more than a biological phenomenon. It is inherently social, psychological, cultural and historical. While this has been recognised by major health actors for decades, open questions remain as to how to build systems that reflect the complexity of health, disease and sickness, and in a context that is increasingly technologised. We argue that an urgent change of approach is necessary. Methods and concepts from the humanities and social science must be embedded in the concepts and methods of the health sciences if we are to promote sustainable interventions capable of engaging with the recognised complexity of health, disease and sickness. Our vision is one of radical interdisciplinarity, integrating aspects of biological, psychological, social and humanities approaches across areas of urgent health need. Radical interdisciplinarity, we argue, entails the practical, methodological and conceptual integration of these approaches to health.

## SIGNATORIES

This text is based on brain writing sessions in which all undersigned took part during the workshop 'Healthy People: How the Medical Humanities and the Health Sciences Study the Complexity of Health' (https://www.lorentzcenter.nl/lc/web/2018/1089/info.php3?wsid=1089&venue=Snellius), held in Leiden between 29 October and 1 November 2018, and supported by: NIAS-Lorentz Center, University of Amsterdam and AMC, University of Bonn and University College London.

Claudi Bockting[a], Giovanni Boniolo[b], Stefano Canali[c], Brendan Clarke[d], Maria Conforti[e], O Caglar Dede[f], Giulia Frezza[e], Mariacarla Gadebusch Bondio[g], Virginia Ghiara[h], Guido Giarelli[i], Phyllis Illari[d], Peng Jia[l], Michael P Kelly[m], Louise Letley[n], Emiliano Mancini[o], Maria Marloth[g] [t], Joko Mulyanto[p], Federica Russo[q], Maartje Schermer[r], Karien Stronks[p], Paolo Vineis[s]

[a] Academic Medical Centre, Department of Psychiatry, University of Amsterdam, Amsterdam, The Netherlands

## Summary

► Socio-psychological-cultural factors are active causes of health.
► Our vision is one of radical interdisciplinarity.
► It is vital to create a joint endeavour across the health sciences, public health, social sciences, and the humanities.

[b] Department of Biomedical Sciences, Università di Ferrara, Ferrara, Italy
[c] Institute for Philosophy, Leibniz University of Hannover, Hannover, Germany
[d] Department of Science and Technology Studies, University College London, London, UK
[e] Department of Molecular Medicine, University of Rome Sapienza, Rome, Italy
[f] Department of Philosophy, Erasmus University of Rotterdam, Rotterdam, The Netherlands
[g] Institute for Medical Humanities, University of Bonn, Bonn, Germany
[h] Department of Philosophy, University of Kent, Canterbury, UK
[i] Faculty of Medicine, University of Catanzaro, Italy
[l] Department of Earth Observation Science, University of Twente, Enschede, The Netherlands
[m] Institute of Public Health, Forvie Site, University of Cambridge, UK
[n] Immunisation, Hepatitis and Blood Safety Department, National Infection Service, Public Health England, London, UK
[o] Institute for Advanced Studies and Computational Science Laboratory, University of Amsterdam, Amsterdam, The Netherlands
[p] Department of Public Health, Academic Medical Center, University of Amsterdam, Amsterdam, The Netherlands
[q] Department of Philosophy, University of Amsterdam, Amsterdam, The Netherlands
[r] Department of Medical Ethics and Philosophy of Medicine, Erasmus MC, Rotterdam, The Netherlands

ˢ School of Public Health, Department of Epidemiology and Biostatistics, Imperial College London, London, UK

ᵗ Department of Psychiatry and Psychotherapy, University Hospital Cologne, Cologne, Germany

## OUR VISION

Health is *more* than the absence of disease. It is also more than a biological phenomenon. It is inherently social, psychological, cultural and historical. Social and personal resources are both key components and key determinants of health, as it has been recognised by major health actors for decades.[1–3] However, open questions remain as to how to build systems that reflect the complexity of health, healthy lives, disease and sickness, and in a context that is increasingly technologised. Although we find in the literature an increasing understanding of the complexity of health,[4–7] the implementation of this knowledge lags behind. Biological approaches to health and disease, as a matter of fact, dominate the development of curative and preventive interventions.

We argue that an urgent change of approach is necessary. Methods and concepts from the humanities and social science must be *embedded* in the concepts and methods of the health sciences and of public health, if we are to promote sustainable interventions capable of engaging with the recognised complexity of health, healthy lives, disease and sickness. This resonates with the vision expressed by the UK Health Secretary and by many policy documents[8 9] from the last decades. Yet, given the difficulties associated with interdisciplinary research, integrated strategies to understand and to intervene on the complexity of health and that engage with biological, social, psychological and behavioural factors are still needed.

Our vision is one of *radical interdisciplinarity*, integrating aspects of biological, psychological, social and humanities approaches across areas of urgent health need. These areas include, but are not confined to, chronic conditions such as the obesity epidemic, cancer and mental health. Radical interdisciplinarity entails the *practical*, *methodological* and *conceptual* integration of approaches to health, as they are developed in the health and social sciences, and in the humanities. It is the combination of cognitive resources from individuals belonging to different disciplines, who accept and respect the division of labour and the resulting epistemic dependence to tackle phenomena that would not be adequately conceptualised within any of the involved discipline alone.[10] In what follows, we describe our current understanding of these three aspects, and describe how radical interdisciplinarity would change them.

## Practical

### How things stand

The impact of social factors on health is widely acknowledged in the scientific literature.[11–15] However, this does not always mean that they are effectively actioned in healthcare and public health. Best practices do exist, but they do not constitute, as yet, the dominant approach.

We think that exemplary public health programmes, such as *WHO Tailoring Immunization Programmes (TIP)*, are strengthened and effective precisely because they explicitly target social factors and exploit deep knowledge of cultural factors and social dynamic in the design of interventions. Despite the successes of interventions targeting social and cultural factors, more distinctively biological models of health predominate in practice.[9] The dominant biological model of disease translates into public health interventions that are based on a simplified logic, according to which to reduce the burden of disease it suffices to reduce exposure. This idea, furthermore, often involves mere behavioural changes at the individual level. While this simplified logic allows for clear, unequivocal criteria of success and avoids complex and difficult discussions about responsibilities at the structural level, it often fails.[16] From the point of view of healthcare systems, in budgets across the high-income countries, this gives rise to the dominance of healthcare spending that aims to cure, rather than prevent, disease. This calls for a deep and interdisciplinary reflection also on what 'care' means—the works of, for example, Mol[17] and Mol *et al*[18] are good examples of how this can be done.

### Our vision for change

Prioritising interdisciplinarity interventions rather than exclusively biological interventions is especially relevant for *prevention, and to obtain sustainable effects of treatment.* Radically interdisciplinary knowledge should then be embedded into practice more widely, for instance, in training of general practitioners, nurses, psychologists and health professionals. This embedding of humanities and social science approaches in public health programmes will gradually be institutionalised and will help solve the major challenges of interdisciplinarity, such as epistemic dependence, the lack of a shared vocabulary and of a mutual recognition of the role played by the other disciplines.[10 19] Existing best practices, such as the 'Tailoring Immunisation programmes (https://www.who.int/immunization/programmes_systems/Global_TIP_overview_July2018.pdf?ua=1)' (TIP), should be exported widely. Yet exporting is hard: how do we know that something that worked once will work in a different context? External validity has long been recognised as crucial in scientific and policy settings,[20] and all the more so once the complexity of health and disease, due to psychosociocultural and historical factors, is brought to the fore. Answering this question requires local and global expertise across the different levels of social organisations. Practitioners make this engagement process possible, while emerging digital technologies can provide new opportunities to involve, as active partners, communities and patients with relevant health experiences.

## Methodological

### How things stand

One diagnosis for the lack of consideration of social and cultural factors in practice is the fragmentation of methods, as they are used in the health and social sciences. While

quantitative methods can be used to effectively study certain health dimensions, psychological and cultural aspects can often be understood only by means of qualitative approaches. How to integrate different epistemological and methodological perspectives, however, is in general not straightforward. For example, obesity has biological, behavioural, psychological, environmental and structural causes.[21 22] Which methods should we use to understand each, and how should we combine them? Different interventions emphasise different causes—biological or social. We might identify structural (sugar tax), environmental (walkable neighbourhoods), behavioural (food labelling) and biological (appetite-modifying drugs) approaches to prevent obesity.[23–25] Yet the existence of different interventions does not produce a unified framework that addresses this plurality of causes, and does not solve the tension between individual and structural responsibilities.[26–28] In addition, innovative biological approaches, such as precision medicine and emerging technologies, are often used as 'magnifier' to see better into 'the biology.'[29 30] Such approaches run the risk of not giving due importance to social and cultural factors influencing patients' life and health.

### Our vision for change

Our plea is for *radical interdisciplinarity*, which places psychosociocultural factors at the heart of health. We integrate methods and concepts used in the health sciences, in the humanities and in the social sciences to understand how each influences and changes our health and our understanding of health. The issue is not just to recognise *that* psychosociocultural factors make a difference to health and disease, but *how and why* they are part of their aetiology, of concept of health itself and of different ways of understanding 'care.' Asking normative questions about health, health interventions and health data use is integral to this approach. This will require reciprocal training for health scientists and practitioners, and those working in the medical humanities and social sciences. In our view, mixed methods research has positively contributed to investigating the complexity of health-related phenomena in an effective way[31 32]; we think that this strategy is likely to be the most suitable to explore the different dimensions of health. Similarly, more integrative methodological approaches are better suited to assess complex dimensions of health interventions such as their impact on health inequalities.[33] To this end, it is useful to think about networks of various kinds of health scientists and how their different methodological practices are integrated. Only by joining forces we can achieve this radical interdisciplinarity and form knowledge that will ground a new generation of methods for studying health and for designing interventions.

### Conceptual

#### How things stand

Just as methods in health are fragmented, so are concepts. We argue that the fragmentation of concepts in the study of health and disease results from the long-standing dominance of biological concepts.[34] But biology alone is insufficient to produce an adequate conceptualisation of health. Similarly, adding a veneer of sociocultural concern to a primarily biological foundation will, we think, not produce an adequate account of the complexity of health issues.

### Our vision for change

Sociopsychological-cultural factors are vital health dimensions and active health *causes*, which, we argue, should be understood on the same footing as biological causes.[35] The radical interdisciplinarity we advocate goes beyond embedding research from the social sciences and humanities into the existing health sciences. Scientific knowledge must be integrated with the lifeworld of individual patients, groups of patients and of populations. This means having deep and detailed understanding of the conditions in which disease develops and evolves, including socioeconomic factors, group dynamic, social support, access to healthcare infrastructure, and so on. To achieve this, we advocate an exercise of thinking and working together to produce conceptual work about health in which different concepts are integrated together from the start: the social sciences and the medical humanities *together* with the health sciences can provide the concepts to account for the complexity of health and disease. This radical interdisciplinarity will produce understandable and practical concepts that can be used in research in the health sciences and in the humanities, and that can be integrated in the design of interventions, for the sake of health.

## HOW TO ACHIEVE THIS VISION

Researchers in the health and social sciences and in the humanities, practitioners and policymakers need to collaborate with a variety of networks of health actors.

To promote health across individuals, integrated approaches to this psychosociocultural view of health require much more than awareness raising. This may involve, for instance, changing the management of health so that the time factor in the relationship between the physician and the patient can be given more importance. We understand the urgent questions at this stage as less about ease of access, but quality of interaction, between health systems and healthy people.

Given the social and group dimension of health, we need to develop measures that engender changes in civil society (families, school, churches, scout, unions). Direct promotion of solidarity, mutual help and healthy lifestyles is unlikely to be effective without careful contributions from the social sciences and humanities. Yet integrated social change can happen in civil society: initiatives to prevent substance abuse in Iceland, for example, were mediated by interventions that strengthen specific community-level factors influencing health, including the creation of a parental network, the promotion of community activities such as parental prowl (parents walking

around their neighbourhood together during weekend nights) and the availability of organised youth activities.[36]

Finally, at the broader level of population, structural interventions that involve transportation, taxation and the fabric of government and commerce all promise potential benefits to health. We cannot hope to develop, evaluate and implement these complex (and usually customised) works without analysing their effects across the many dimensions of practice and the social lifeworld.

Radical interdisciplinarity faces challenges at all these levels. First successful attempts have been applied using an integrated approach (for example, a dynamical systems approach to explain transmission of resistance in HIV spreading and in-stent restenosis in coronary artery disease.[37] We are aware of the scale of these challenges, but we are also confident that a new course in the way we study and intervene on health is achievable. This manifesto, a joint endeavour across the health sciences, public health, social sciences and the humanities, is itself an exercise in radical interdisciplinarity, based on combined efforts to create a common ground for the future of research on health and disease, and for the design and implementation of interventions that can drive healthcare systems towards health.

**Contributors** BC, VG and FR drafted a full text on the basis of brain writing sessions which all signatories of this communication article took part in during the workshop 'Healthy People: How the Medical Humanities and the Health Sciences Study the Complexity of Health,' held in Leiden between 29 October and 1 November 2018.

**Funding** This study received funding from the NIAS-Lorentz Center, University of Amsterdam and AMC, University of Bonn and University College London that made possible the organisation of the workshop 'Healthy People: How the Medical Humanities and the Health Sciences Study the Complexity of Health.'

**Competing interests** None declared.

**Patient consent for publication** Not required.

**Provenance and peer review** Not commissioned; externally peer reviewed.

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
