## [Reviewer comments · BMJ Open]

ARTICLE DETAILS

TITLE (PROVISIONAL)	Time to care: why the humanities and the social sciences belong in the science of health
AUTHORS	Clarke, B; Ghiara, Virginia; Russo, Federica

VERSION 1 - REVIEW

REVIEWER	STEPHEN HINCHLIFFE University of Exeter, UK
REVIEW RETURNED	11-Apr-2019

GENERAL COMMENTS	Time to care While I support the central argument, that there is need for culturally and socially informed approaches to health, the paper is really no more than a familiar (and surely commonplace) statement to this effect. In re-stating the case, there is little or no appreciation of the issues involved in overcoming barriers to improvement or indeed the real hurdles that need to be cleared if we are really to deliver “radical interdisciplinarity” (which is undefined) and health improvements. Only by stating and appreciating the barriers can we really hope to overcome the current limitations in approaches to health. I will limit myself to 4 points that are illustrative of the requirements for a much-improved argument. 1. Health – health is not the absence of disease. The authors nearly say as much but then focus on the importance of social and cultural factors to disease prevention. But health itself is a contested concept see https://www.nature.com/articles/s41599-018-0113-9 and most people live with some kind of disease in ways that are more or less healthy. Indeed, the paradigmatic shifts from biomedical approaches to disease and the moves towards health involve much more than changing aetiological accounts. They move us into a space where there is an active debate concerning what constitutes healthy lives, both in terms of individuals and across social groups.2. Interdisciplinarity – it is naïve to suggest that integrating methods and concepts is either enough or indeed a simple matter of bringing things together. Some acknowledgement at least of the difficulties of doing interdisciplinarity is required (see for example Callard F, Fitzgerald D (2015) Rethinking interdisciplinarity across
--

	the social sciences and neurosciences. Palgrave Macmillan, UK). If it was easy, as they say, word would have got out. 3. The policy and practical difficulties of implementing this approach really need to be stated if readers of BMJ are going to do anything more than shrug their shoulders. Why, given all the shortfalls, are biomedical approaches so successful? Why do policy-makers and health care professionals find social and cultural research knowledge so difficult to use? Why for example do they often resist anything that might be construed as social and also 'political' (obesity is a lot easier in Whitehall if you blame individuals and / or seek biomedical interventions – it's harder if you have to confront the industrial food complex!) How do you integrate often qualitative insights into cultural dynamics with the largely quantitative analysis of epidemiologists and biomedical researchers? How do health care delivery models (like for example the reduction of consultation times) militate against anything but reductive and disease-oriented practice? Indeed, given the explosion of writing on care in recent years, how does the title of this piece engage with contested accounts of what it means to do care? (see AM Mol, Logics of Care). Finally, and more positively, the authors mention technology (though say nothing very useful about it). I presume they, in part at least, mean the data-rich nature of health care – how might this allow us to bring social and cultural knowledge to the clinic? Or does it further sediment a biomedical view? 4. The micro-, meso etc levels argument feels very passé – health care and delivery as well as cultural and social accounts have long since moved on from a Russian doll version of service – it is more useful to think about networks of various kinds of health actors. In sum, I would support such a communication piece in the journal, but this is a fair distance from what I think readers would expect (and the kinds of arguments with which they could meaningfully engage) and is not what I would suggest is the current state of play in social science, medical humanities.
--	---

REVIEWER	Aurea Maria Zöllner Ianni Faculty of Public Health, University of São Paulo, Brazil.
REVIEW RETURNED	28-Apr-2019

GENERAL COMMENTS	The article is clear and objective. The authors say that this paper is a manifesto claiming that "a joint endeavour across the health sciences, public health, social sciences, and the humanities" is a radical interdisciplinarity exercise. The issues raised in it can be interesting specially for researchers, policy makers, health professionals, and doctors of all disciplines. Although it has no innovative ideas exactly, they must be considered in the public health and medicine areas, in the sense that the questions that are in it are important not only for nowadays but specially to the future. They present clearly the practical, methodological and conceptual questions involved in seeing health and diseases, sickness and illness beyond biology. This proposal is very didactic and provocative.
---

VERSION 1 – AUTHOR RESPONSE

Reviewer1

While I support the central argument, that there is need for culturally and socially informed approaches to health, the paper is really no more than a familiar (and surely commonplace) statement to this effect. In re-stating the case, there is little or no appreciation of the issues involved in overcoming barriers to improvement or indeed the real hurdles that need to be cleared if we are really to deliver “radical interdisciplinarity” (which is undefined) and health improvements. Only by stating and appreciating the barriers can we really hope to overcome the current limitations in approaches to health. I will limit myself to 4 points that are illustrative of the requirements for a much-improved argument.

Response: We understand the reviewer may feel we haven’t really contributed to overcoming the existing barriers in understanding and promoting health. We never intended, in such a short text, to give the impression that this can be easily done. Our contribution is to state why the problem exist, distinguishing practical, methodological, and conceptual aspects, and to point to the way forward – radical interdisciplinarity – which remains programmatic at this stage. We tried to clarify the points raised in the comments below, keeping added text to a minimum. Should the editors allow us for more space, we can certainly elaborate more on these aspects.

Comment 1:

Health – health is not the absence of disease. The authors nearly say as much but then focus on the importance of social and cultural factors to disease prevention. But health itself is a contested concept see <https://www.nature.com/articles/s41599-018-0113-9> and most people live with some kind of disease in ways that are more or less healthy. Indeed, the paradigmatic shifts from biomedical approaches to disease and the moves towards health involve much more than changing aetiological accounts. They move us into a space where there is an active debate concerning what constitutes healthy lives, both in terms of individuals and across social groups. 
Response: We appreciate your comments very much. As you recommended, we have specified that health is not the mere absence of disease, and that the complexity of health entails complex debates on what “healthy lives” are. We also acknowledged that social and cultural factors are not only important determinants of health, but also vital components of the concept of health.

Changes: We changed the first sentence of the manuscript to include that health is more than the absence of disease.

We also added the sentence:

“Social and personal resources are both key components and key determinants of health, as it has been recognised by major health actors for decades (Hinchliffe et al., 2018; Huber et al., 2011; WHO, 2006)” (p. 2).

We changed the following sentence to add the concept of healthy lives:

“However, open questions remain as to how to build systems that reflect the complexity of health, healthy lives, disease, and sickness, and in a context that is increasingly technologized” (p. 2).

Comment 2:

Interdisciplinarity – it is naïve to suggest that integrating methods and concepts is either enough or indeed a simple matter of bringing things together. Some acknowledgement at least of the difficulties of doing interdisciplinarity is required (see for example Callard F, Fitzgerald D (2015) Rethinking interdisciplinarity across the social sciences and neurosciences. Palgrave Macmillan, UK). If it was easy, as they say, word would have got out.

Response: We agree that interdisciplinarity is not merely the integration of methods and concepts. As we explained on page 6, this manifesto is itself the result of a long interdisciplinary discussion that required time and efforts. And it is but the beginning of joint efforts between different disciplinary perspectives. Our proposal is not based on the idea that interdisciplinary works are easy, rather we propose some possible interventions to make interdisciplinarity easier in the future, such as embedding the humanities and social science in training of GPs, nurses, psychologists and health professionals. We agree, nevertheless, that our manuscript can clarify some of challenges associated with interdisciplinarity.

Changes: We added a sentence to define radically interdisciplinarity and one sentence to acknowledge the challenges associated with interdisciplinarity. We also clarified that the manifesto is itself an exercise in radical interdisciplinarity:

“[radically interdisciplinarity] is the combination of cognitive resources from individuals belonging to different disciplines, who accept and respect the division of labour and the resulting epistemic dependence to tackle phenomena that would not be adequately conceptualised within any of the involved discipline alone (Andersen, 2016)” (p. 3).

“This embedding of humanities and social science approaches in public health programmes will gradually be institutionalised and will help to solve the major challenges of interdisciplinarity, such as epistemic dependence, the lack of a shared vocabulary and of a mutual recognition of the role played by the other disciplines (Andersen, 2016; Callard & Fitzgerald, 2015)” (p. 4).

“This manifesto, a joint endeavour across the health sciences, public health, social sciences, and the humanities, is itself an exercise in radical interdisciplinarity, based on combined efforts to create a common ground for the future of research on health and disease, and for design and implementation interventions that can drive health care systems towards health” (p. 7).

Comment 3:

The policy and practical difficulties of implementing this approach really need to be stated if readers of BMJ are going to do anything more than shrug their shoulders. Why, given all the shortfalls, are biomedical approaches so successful? Why do policy-makers and health care professionals find social and cultural research knowledge so difficult to use? Why for example do they often resist anything that might be construed as social and also ‘political’ (obesity is a lot easier in Whitehall if you blame individuals and / or seek biomedical interventions – it’s harder if you have to confront the industrial food complex!) How do you integrate often qualitative insights into cultural dynamics with the largely quantitative analysis of epidemiologists and biomedical researchers? How do health care delivery models (like for example the reduction of consultation times) militate against anything but reductive and disease-oriented practice? Indeed, given the explosion of writing on care in recent years, how does the title of this piece engage with contested accounts of what it means to do care? (see AM Mol, Logics of Care). Finally, and more positively, the authors mention technology (though say nothing very useful about it). I presume they, in part at least, mean the data-rich nature of health care – how might this allow us to bring social and cultural knowledge to the clinic? Or does it further sediment a biomedical view?

Response: Most of the questions asked by the reviewer are unfortunately too complex to be discussed in detail in this manuscript, which was originally allowed only 2,500 words. With long-term projects that practice this ‘radical interdisciplinarity’, we hope to address precisely these questions, and in the complexity mentioned by the reviewer. We have, however, added short claims and related references to clarify our points.

We agree that the contribution of Mol and of Mol, Moser, Pols are very good example of how to question the very meaning of ‘care’, and that indeed is aligned with our vision.

Concerning the simplified logic, according to which to reduce the burden of disease it suffices to reduce exposure, our claim is that this logic allows for the development of straightforward programmes that can be evaluated in a clear and unequivocal way.

As for the different methodological perspectives, over the last decades good examples of mixed methods studies have shown that their integration is possible.

Concerning structural and individual responsibilities, we have briefly introduced this problem arguing against the reduction of health to individual behaviours.

Technologies, we argue, can offer new opportunities to engage with patients and communities, but at the same time cause the current health data deluge, which might increase the emphasis on the biology of health. The use of such technologies and health data, furthermore, requires ethical and normative discussions.

Changes:

To consider the important contribution of Mol and of Mol, Moser and Pols, we added the sentence:

“This calls for a deep and interdisciplinary reflection also on what ‘care’ means – the work of e.g. Mol (2008) and Mol, Mosel, Pols (2010) are good examples of how this can be done” (p. 4).

To explain briefly why biomedical, simplified models are successful, we added the sentence:

“While this simplified logic allows for clear, unequivocal criteria of success and avoids complex and difficult discussions about responsibilities at the structural level, it often fails (Kelly & Russo, 2017)” (p. 4).

To consider the challenges and solutions associated with the methodological differences between disciplines, we added the sentences:

“While quantitative methods can be used to effectively study certain health dimensions, psychological and cultural aspects can often be understood only by means of qualitative approaches. How to integrate different epistemological and methodological perspectives, however, is in general not straightforward” (p. 5).

“In our view, mixed methods research has positively contributed to investigating the complexity of health-related phenomena in an effective way (Greene, 2015; Tariq & Woodman, 2013); we think that this strategy is likely to be the most suitable to explore the different dimensions of health. Similarly, more integrative methodological approaches are better suited to assess complex dimensions of health interventions such as their impact on health inequalities (Dede, 2019). To this end, it is useful to think about networks of various kinds of health scientists and how their different methodological practices are integrated.” (p. 5-6).

To consider the role played by both structural and individual health behaviours, we added three sentences:

“The dominant biological model of disease translates into public health interventions that are based on a simplified logic, according to which to reduce the burden of disease it suffices to reduce exposure. This idea, furthermore, often involves mere behavioural changes at the individual level. While this simplified logic allows for clear, unequivocal criteria of success and avoids complex and difficult discussions about responsibilities at the structural level, it often fails (Kelly & Russo, 2017)” (p. 4).

“Yet the existence of different interventions does not produce a unified framework that addresses this plurality of causes, and does not solve the tension between individual and structural responsibilities (Powers & Faden, 2008; Stol, Schermer, & Asscher, 2016; Venkatapuram, 2011)” (p. 5).

To discuss the role played by emerging health technologies, we added three sentences:

“emerging digital technologies can provide new opportunities to involve, as active partners, communities and patients with relevant health experiences” (p. 4).

“In addition, innovative biological approaches, such as precision medicine and emerging technologies are often used as ‘magnifier’ to see better into ‘the biology’ (Jia, 2019; Saracci, 2018). Such approaches run the risk of not giving due importance to social and cultural factors influencing patients’ life and health” (p. 5).

“Asking normative questions about health, health interventions, and health data use is integral to this approach” (p. 5).

Comment 4:

The micro-, meso etc levels argument feels very passé – health care and delivery as well as cultural and social accounts have long since moved on from a Russian doll version of service – it is more useful to think about networks of various kinds of health actors.

Response: We thank the reviewer for the observation. We decided to avoid talking about micro-, meso- and macro- levels.

In sum, I would support such a communication piece in the journal, but this is a fair distance from what I think readers would expect (and the kinds of arguments with which they could meaningfully

engage) and is not what I would suggest is the current state of play in social science, medical humanities.

Response: We acknowledge the complexity of the topic. Given that communication pieces are meant to be short, the aim is not to provide an exhaustive discussion on how to solve the barriers to improvement. Rather, our aim is to express clearly what challenges should be solved and to point the way ahead in a pragmatic way. This has been captured by Reviewer2, when they say that "Although it has no innovative ideas exactly, they must be considered in the public health and medicine areas, in the sense that the questions that are in it are important not only for nowadays but specially to the future".

Reviewer2

The article is clear and objective. The authors say that this paper is a manifesto claiming that "a joint endeavour across the health sciences, public health, social sciences, and the humanities" is a radical interdisciplinarity exercise. The issues raised in it can be interesting specially for researchers, policy makers, health professionals, and doctors of all disciplines. Although it has no innovative ideas exactly, they must be considered in the public health and medicine areas, in the sense that the questions that are in it are important not only for nowadays but specially to the future. They present clearly the practical, methodological and conceptual questions involved in seeing health and diseases, sickness and illness beyond biology. This proposal is very didactic and provocative.

Response: Thank you for the positive comments. We are pleased to see that the spirit with which this text is written came across clearly and received positively by the reviewer.

VERSION 2 – REVIEW

REVIEWER	S Hinchliffe University of Exeter
REVIEW RETURNED	04-Jul-2019

GENERAL COMMENTS	Thanks you for addressing the concerns raised in the previous copy. The resulting paper is more convincing and makes a valid contribution to the literature and debate.
---